# The Synergistic Effect of Co-Treatment of Methyl Jasmonate and Cyclodextrins on Pterocarpan Production in *Sophora flavescens* Cell Cultures

**DOI:** 10.3390/ijms21113944

**Published:** 2020-05-30

**Authors:** Soyoung Kim, Yu Jeong Jeong, Su Hyun Park, Sung-Chul Park, Saet Buyl Lee, Jiyoung Lee, Suk Weon Kim, Bo-Keun Ha, Hyun-Soon Kim, HyeRan Kim, Young Bae Ryu, Jae Cheol Jeong, Cha Young Kim

**Affiliations:** 1Biological Resource Center, Korea Research Institute of Bioscience and Biotechnology (KRIBB), Jeongeup 56212, Korea; cherish767@kribb.re.kr (S.K.); yjjeong@kribb.re.kr (Y.J.J.); suhyun@kribb.re.kr (S.H.P.); heypsc@kribb.re.kr (S.-C.P.); sblee@kribb.re.kr (S.B.L.); jiyoung1@kribb.re.kr (J.L.); kimsw@kribb.re.kr (S.W.K.); 2Department of Plant Biotechnology, College of Agriculture and Life Science, Chonnam National University, Gwangju 61186, Korea; bkha@jnu.ac.kr; 3Plant Systems Engineering Research Center, Korea Research Institute of Bioscience and Biotechnology (KRIBB), Daejeon 34141, Korea; hyuns@kribb.re.kr (H.-S.K.); kimhr@kribb.re.kr (H.K.); 4Functional Biomaterials Research Center, Korea Research Institute of Bioscience and Biotechnology (KRIBB), Jeongeup 56212, Korea; ybryu@kribb.re.kr

**Keywords:** *Sophora flavescens*, cell cultures, pterocarpan, maackiain, trifolirhizin, elicitation

## Abstract

Pterocarpans are derivatives of isoflavonoids, found in many species of the family Fabaceae. *Sophora flavescens* Aiton is a promising traditional Asian medicinal plant. Plant cell suspension cultures represent an excellent source for the production of valuable secondary metabolites. Herein, we found that methyl jasmonate (MJ) elicited the activation of pterocarpan biosynthetic genes in cell suspension cultures of *S. flavescens* and enhanced the accumulation of pterocarpans, producing mainly trifolirhizin, trifolirhizin malonate, and maackiain. MJ application stimulated the expression of structural genes (*PAL, C4H, 4CL, CHS, CHR, CHI, IFS, I3’H,* and *IFR*) of the pterocarpan biosynthetic pathway. In addition, the co-treatment of MJ and methyl-β-cyclodextrin (MeβCD) as a solubilizer exhibited a synergistic effect on the activation of the pterocarpan biosynthetic genes. The maximum level of total pterocarpan production (37.2 mg/g dry weight (DW)) was obtained on day 17 after the application of 50 μM MJ on cells. We also found that the combined treatment of cells for seven days with MJ and MeβCD synergistically induced the pterocarpan production (trifolirhizin, trifolirhizin malonate, and maackiain) in the cells (58 mg/g DW) and culture medium (222.7 mg/L). Noteworthy, the co-treatment only stimulated the elevated extracellular production of maackiain in the culture medium, indicating its extracellular secretion; however, its glycosides (trifolirhizin and trifolirhizin malonate) were not detected in any significant amounts in the culture medium. This work provides new strategies for the pterocarpan production in plant cell suspension cultures, and shows MeβCD to be an effective solubilizer for the extracellular production of maackiain in the cell cultures of *S. flavescens*.

## 1. Introduction

Plants produce a wide variety of secondary metabolites that are economically important as pharmaceuticals, agrochemicals, bio-pesticides, flavors, fragrances, colors, and food additives [1,2]. *Sophora flavescens* Aiton, the shrubby sophora, belongs to the family Fabaceae (Leguminosae) and is an important legume in Asia [3]. The root of *S. flavescens*, also known as Kosam in Korean (Kushen in Chinese), is commonly used as a traditional herbal medicine because of its various biological activities, including anti-cancer, anti-arrhythmic, anti-allergic, anti-inflammatory, and anti-asthmatic effects [3,4,5,6]. It has been officially listed in the Korean Pharmacopoeia and Chinese Pharmacopoeia. *S. flavescens* is known to produce a wide range of secondary metabolites, including flavonoids, alkaloids, triterpenoids, and pterocarpans [3]. Among them, pterocarpans—including maackiain, trifolirhizin, and trifolirhizin malonate—are isoflavonoids found in many species of Fabaceae. They have been reported to have various biological activities, such as anti-microbial, anti-cancerous, anti-inflammatory, and anti-malarial activities [7,8]. Pterocarpans comprise the second largest group of isoflavonoids after isoflavones. They primarily act as phytoalexins in leguminous plants, and also function as signal molecules in plant–microorganism interactions [7,9]. Leguminous plants are known to produce pterocarpan phytoalexins [10]. Pterocarpans of pisatin, medicarpin, and maackiain are produced in pea (*Pisum sativum*), peanut (*Arachis hypogaea*), and *Sophora japonica*, respectively [10,11,12,13]. Pterocarpans are formed at the last stages of the flavonoid biosynthetic pathway as a part of the isoflavonoid branch [7]. Isoflavonoids are synthesized by a legume-specific branch of the general phenylpropanoid pathway. The first product in the biosynthesis of flavonoids and isoflavonoids is chalcone, which requires an enzymatic reaction catalyzed by chalcone synthase (CHS) [14]. Isoliquiritigenin and liquiritigenin are produced from chalcones by additional legume-specific enzymes, chalcone reductase (CHR) and chalcone isomerase (CHI) [15,16]. Isoflavonoid biosynthesis is further catalyzed by isoflavone synthase (IFS) to biosynthesize isoflavanone (daidzein), which is then modified by methyltransferase (I4’OMT), hydroxylase (I3’H), reductase (IFR), glucosyltransferase (IF7GT), and malonyltransferase (IF7MaT) to form isoflavonoids (pterocarpans) (Figure 1) [17]. The flavonoid biosynthetic pathway has been elucidated, and many of the related genes have been characterized in other species [18]. Recently, putative genes involved in the flavonoid biosynthesis in *S. flavescens* were discovered by transcriptome (RNA-Seq) analysis [5,19]. Lee et al. [5] analyzed the contents of phenolic compounds in the leaves, stems, and roots of *S. flavescens*, and have identified candidate genes involved in the biosynthesis of phenolic acids and flavonoids. They have also reported the expression patterns of the biosynthesis-related genes in different organs and developmental stages.

Plant cell cultures have been perceived as attractive sources for the production of biologically active secondary metabolites under controlled conditions independent of climatic changes or soil conditions [20,21,22,23,24]. Another effective method involves elicitation with various biotic and abiotic elicitors to enhance the production of secondary metabolites in plant cell cultures [21,23,25]. Cyclodextrins have been widely reported as an effective solubilizer for poorly aqueous soluble drug [26,27]. They are a family of cyclic oligosaccharides composed of six, seven, or eight glucopyranose units (referred to as α-, β-, or γ-cyclodextrins, respectively). Methylated-β-cyclodextrin (MeβCD) is most commonly used to enhance the solubility of poorly water-soluble chemical compounds [17,28,29]. In an earlier study, combined elicitation with methyl jasmonate (MJ) and MeβCD was shown to be the most efficient method for the extracellular production of resveratrol and its derivatives in grapevine cell cultures [29].

Biotechnological production of secondary metabolites in plant cell cultures offers an attractive alternative to the extraction of whole plant material. So far, more than 200 compounds, including ones with flavonoids and alkaloids as the major components, have been isolated from *S. flavescens* [30]. Furuya and Ikuta [30] were the first to establish callus cultures of *S. flavescens*, and they found that the callus produced (*R*)-maackiain and pterocarpan. Yamamoto et al. [31,32,33] also reported that callus cultures of *S. favescens* produced both prenylated flavanones (sophoraflavanone G and lehmannin) and pterocarpans [31,32,33]. They also reported that maackiain was not found in the fresh roots, but accumulated in the callus cultures of *S. flavescens* as the 6′-malonyl ester of its 3-*O*-glucoside (trifolirhizin malonate) [31].

In this study, we demonstrate that the application of MJ alone or together with MeβCD in the cell cultures of *S. flavescens* has a significant and synergistic effect on the expression of structural genes involved in pterocarpan biosynthesis, thereby leading to an enhanced production of pterocarpans, such as maackiain, trifolirhizin, and trifolirhizin malonate. We further report that combined treatment with MJ and MeβCD synergistically enhances the production of pterocarpans and preferentially induces the secretion of maackiain into the cell culture medium.

## 2. Results and Discussion

### 2.1. Establishment and Optimization of the Cell Suspension Cultures of S. flavescens

The cell suspension cultures of *S. flavescens* were established using the leaf-derived calli (line no. KS611, BP1429378), which had been determined as the cell line of greatest potential (callus growth) during cell line selection. The growth of the cell suspension cultures in MS1B2P medium was measured to determine the most suitable growth time point for cell harvesting. The growth curves of the cell suspension cultures treated with or without MJ were monitored by simply weighing the suspension cells (dry weight, DW) harvested at each time point. The cell suspension cultures of *S. flavescens* were monitored at 2 day intervals for 16 days (Figure 2). The cell suspension cultures exhibited a sigmoid growth pattern similar to those of other plant cell suspension cultures [23,34]. The pattern of the growth curve showed an initial lag phase during the first 8 days, followed by a period of exponential growth until day 14. After the exponential phase, cell cultures entered the stationary phase when cells showed a decrease in cell density. The MJ-treated cells showed a similar growth pattern to that in the untreated cells, but they displayed about 38% reduction in cell biomass on day 12 compared to the untreated cells (Figure 2A). The highest biomass of the *S. flavescens* cell cultures was obtained on day 14 in both untreated (16.1 g/L) and MJ-treated (16.3 g/L) cells. As shown in Figure 2B, microscopic observations of *S. flavescens* cell cultures exhibited cell morphology with non-embryogenic characteristics, such as irregular, longish, and multicellular aggregates in shapes [23,35,36].

### 2.2. Effects of Various Elicitors on Pterocarpan Accumulation in Cell Suspension Cultures of S. flavescens

To investigate the effects of various elicitors on pterocarpan biosynthesis in cell suspension cultures of *S. flavescens*, we analyzed the accumulation of pterocarpans by HPLC, following the elicitation of the cells with methyl jasmonate (MJ), methyl viologen (MV), salicylic acid (SA), chitosan (Chi), ethephon (ET), and ultraviolet (UV) light. The total content of pterocarpans was significantly induced in the 100 μM, MJ-treated suspension cells (~3.5-fold), while ET induced the accumulation of pterocarpans slightly (~2.1-fold). However, no significant increase was observed in the control and other treated cells (Figure 3). As per the HPLC profiles, MJ treatment led to the most significant increase in peaks 1, 2, and 3, corresponding to trifolirhizin, trifolirhizin malonate, and maackiain, respectively. In this experiment, the two peaks (1 and 3) eluting at 7.88 min and 14.28 min were identified by comparison with the trifolirhizin and maackiain standards via HPLC analysis. The peak (2) at the retention time 8.93 min was further analyzed and identified as trifolirhizin malonate by the recycling preparative HPLC and NMR analysis (Appendix A). When compared to the control, the maximum levels for the total pterocarpan contents (maackiain, trifolirhizin, and trifolirhizin malonate) were elevated nearly 3.5-fold in the MJ-treated suspension cells, representing 3.4-fold, 3.6-fold, and 3.6-fold increase in maackiain, trifolirhizin, and trifolirhizin malonate, respectively. We noticed that the purified trifolirhizin malonate in 80% methanol solution was easily converted into trifolirhizin during its storage at room temperature (data not shown). Likewise, Yamamoto et al. [31] also demonstrated that trifolirhizin malonate readily changed into trifolirhizin during the Soxhlet extraction. They found that trifolirhizin malonate dissolved in methanol was quantitatively converted to trifolirhizin by reflux within 15 h. By contrast, maackiain was not detected after refluxing for 15 h. Together, these results indicate that trifolirhizin malonate is likely unstable and is changeable to trifolirhizin.

### 2.3. Effects of Methyl Jasmonate on Pterocarpan Contents in Cell Suspension Cultures of S. flavescens

Previous studies have reported that MJ is a potent elicitor to enhance secondary metabolites production [23,37,38,39,40]. We have also reported that the biosynthesis of isoflavonoids is induced in MJ-treated soybean suspension cells [23]. Thus, to obtain the optimal elicitation conditions for pterocarpan production in cell suspension cultures of *S. flavescens,* the pterocarpan contents were determined by HPLC following an MJ elicitation. As shown in Figure 4, the pterocarpan production in the *S. flavescens* cells treated with MJ (20 μM to 100 μM) for 3 days increased significantly compared with that in the control (about 3-fold), suggesting that 50 μM MJ is the optimal elicitation concentration for an enhanced production of pterocarpans in the cell suspension cultures of *S. flavescens* (Figure 4A). Trifolirhizin malonate accumulated as a major compound (more than 70%) of the maackiain derivatives by MJ elicitation. The production of pterocarpans by the *S. flavescens* cells increased slightly as the MJ concentration increased, up to 50 μM, whereas a slight decrease of pterocarpan production was observed after the treatment of cells with 60 μM and 100 μM MJ. The highest content of pterocarpans was around 9.41 mg/g DW at 3 days following the elicitation of cells with 50 μM MJ. To further obtain the appropriate time point for high-level production of pterocarpans in the 50 μM MJ-treated cell cultures, we monitored the total pterocarpan production at different time points until 28 days after elicitation (Figure 4B and Table 1). During this process, total pterocarpan contents increased significantly until 17 days after elicitation, and the maximum content of 31.4 mg/g DW to 37.2 mg/g DW, which corresponded to approximately 11.7-fold to 13.8-fold increases, respectively, was observed between 10 and 17 days after elicitation. However, the pterocarpan production in untreated cells only slightly increased until 28 days (2.0 mg/g DW to 2.8 mg/g DW).

These results represent that MJ elicitation promotes the accumulation of pterocarpans, including trifolirhizin malonate as a major compound, in *S. flavescens* cell cultures. Weidemann et al. [41] had previously reported that maximum production of maackiain and trifolirhizin malonate in chickpea cell cultures treated with yeast extract were approximately 8.5 μg/g fresh weight (FW) and 159.7 μg/g FW, respectively. This suggests that MJ can be used an effective elicitor to produce high levels of pterocarpans in the cell cultures of leguminous plants.

### 2.4. Transcriptional Activation of the Pterocarpan Biosynthetic Genes by Methyl Jasmonate and Methyl-β-Cyclodextrin Elicitation

We have previously reported the identification and expression analysis of the structural genes involved in the biosynthesis of phenolic compounds, including isoflavonoids, in different organs and developmental stages of *S. flavescens* [5]. Likewise, Jiao et al. [42] also performed the genome-wide analysis of structural genes involved in legume-specific isoflavonoid biosynthesis, such as *SfCHS, SfCHR, SfIFS,* and *SfIFR* in *S. flavescens* [42]. Pterocarpans are synthesized from L-phenylalanine through the general phenylpropanoid pathway and the isoflavonoid biosynthetic pathway via a series of enzymes (Figure 1). In leguminous plants, CHS, CHR, CHI, IFS, and IFR are key enzymes involved in the isoflavonoid biosynthesis. Thus, to analyze the relationship between pterocarpan accumulation and expression of pterocarpan biosynthetic genes in the *S. flavescens* cells, elicited with MJ alone or together with MeβCD, qRT-PCR was performed to examine the expression of structural genes involved in the pterocarpan biosynthetic pathway (Figure 5). Among them, biosynthetic genes, such as *SfPAL, SfC4H, Sf4CL, SfCHS, SfCHR, SfCHI, SfIFS, SfIF3’H,* and *SfIFR* were up-regulated by the 50 μM MJ or 50 mM MeβCD elicitation of cells for 24 h. In addition, the co-treatment with MJ and MeβCD synergistically induced the expression of the pterocarpan biosynthetic genes. The activation of the three genes *SfPAL, SfC4H,* and *Sf4CL*, encoding PAL, C4H, and 4CL, respectively, in the phenylpropanoid pathway likely provides more precursors for flavonoid biosynthesis. Furthermore, the *SfCHR3* gene was most highly induced by MJ alone (~8.1-fold) or by co-treatment with MeβCD (~41.0-fold) relative to the control. CHR is the key enzyme that catalyzes the first committed step in isoliquiritigenin biosynthesis [42,43]. This may be the reason for the high responsiveness of the *SfCHR3* gene of *S. flavescens* cells to MJ and MJ + MeβCD. Increased CHR enzyme activity can further facilitate the production of the isoliquiritigenin precursor for isoflavonoid biosynthesis. The *SfIF7GT* gene was not significantly responded to with MJ and MJ + MeβCD treatments, but its basal level was high, being approximately 7.9-fold higher than that of the *SfACT11* gene used as an internal control (Appendix A). Such a high-level expression of the IF7GT enzyme may result in the accumulation of trifolirhizin through the catalysis of maackiain glycosylation. These results thus indicate that MJ alone or co-treatment with MeβCD induces the transcriptional activation of multiple structural genes involved in pterocarpan biosynthesis, thereby leading to an enhanced pterocarpan production in *S. flavescens* cells.

As shown in Figure 5, MeβCD application alone also induced the expression of the pterocarpan biosynthetic genes in *S. flavescens* cell cultures, indicating that MeβCD is an effective elicitor. Previous studies have also reported that the addition of MeβCD to cell cultures activate the structural genes in the metabolic pathways, as well as increase the extracellular production of secondary metabolites [44,45,46]. Likely, our results demonstrate that MeβCD acts not only as an elicitor for the expression of pterocarpan biosynthetic genes, but also as a solubilizer for the enhanced production of pterocarpans.

### 2.5. Cyclodextrin Effectively Induces Maackiain Production in Culture Medium of the MJ-treated Cells

MeβCD is most widely used as an effective solubilizing agent for an enhanced extracellular production of the secondary metabolites in the cell cultures [29]. We also found that the co-treatment with MJ and MeβCD had a synergistic effect on the activation of multiple pterocarpan biosynthetic genes in *S. flavescens* cells. Thus, we investigated the effects of MeβCD in combination with MJ on the extracellular production of pterocarpans in cell cultures of *S. flavescens*. As shown in Figure 6 and Table 2, 50 μM MJ treatment alone induced 3.5-fold, 3.8-fold, and 2.3-fold pterocarpan production in trifolirhizin (1), trifolirhizin malonate (2), and maackiain (3), respectively (22.4 mg/g DW) in the *S. flavescens* cells, but was not detected in the culture medium. Furthermore, 50 mM MeβCD alone also moderately increased the accumulation of pterocarpans (1, 2, and 3) in the cells (10.9 mg/g DW) and only maackiain (3) in the medium (30.4 mg/L). As hypothesized, the combined applications of cells with MJ and MeβCD synergistically induced a high production of pterocarpans in the cells (9.4-fold, 9.6-fold, and 7.8-fold in 1, 2, and 3, respectively; 58.0 mg/g DW). Notably, we found that among the pterocarpans, only maackiain considerably accumulated in the culture medium after MJ and MeβCD treatments (222.7 mg/L), indicating its extracellular secretion due to MeβCD (Table 2). Therefore, MeβCD facilitates the extracellular production of maackiain (aglycone) into the culture medium. Structurally, MeβCD has a hydrophobic central pocket with a hydrophilic outer surface, and is well known to be capable of forming an inclusion complex with a small molecule and thus enhance its complex solubility [26,27]. The aglycone maackiain likely fits well into the hydrophobic central pocket of the MeβCD structure, thereby forming a cyclodextrin complex with maackiain. These cyclodextrin/maackiain complexes may help to increase MeβCD’s solubility in the culture medium. However, MeβCD likely does not form a soluble complex with other pterocarpans (trifolirhizin and trifolirhizin malonate).

To conclude, we demonstrated the enhancement of pterocarpan production, including maackiain, trifolirhizin, and trifolirhizin malonate, in the *S. flavescens* cell cultures through the elicitor-mediated method with MJ and MeβCD. This resulted due to the transcriptional activation of the pterocarpan biosynthetic genes during MJ and MeβCD elicitation. Furthermore, we found that the combined treatment of cells with MJ and MeβCD not only had synergistic effects on pterocarpan production, but also on the extracellular production of maackiain into the culture medium. This work, therefore, indicates that the elicitor-mediated approach, along with a solubilizer, such as MeβCD, can be potentially used to enhance the production of secondary metabolites in other plant cell cultures.

## 3. Materials and Methods

### 3.1. Callus Induction and Cell Cultures

The *S. flavescens* calli used in this study were induced from the leaf of *S. flavescens*, and the calli were maintained at 24 °C in the dark in Murashige and Skoog (MS) medium (pH 5.8), supplemented with 30 g/L sucrose, 1.0 mg/L 6-benzylaminopurine (BAP), and 2.0 mg/L picloram (designated MS1B2P medium). The MS1B2P medium was solidified with 4 g/L Gelrite (Duchefa). The best calli were deposited at the Korean Collection for Type Cultures (KCTC; http://bioproduct.kribb.re.kr) as bio-product BP1429378. Cell cultures were established by transferring homogeneous callus to liquid medium, and were subcultured at two-week intervals. The cultured cells (~5 g fresh weight (FW), 10% (*v/v*) inoculum) were transferred into 50 mL of MS1B2P liquid medium in 250 mL flasks and cultured on a rotary shaker at 90 rpm and 24 °C in the dark.

### 3.2. Elicitation of the S. flavescens Cell Cultures

Cell cultures were subcultured into 50 mL of MS1B2P liquid medium in 250 mL flasks and placed on a rotary shaker at 90 rpm and 24 °C in the dark for 7 days, as previously described [23]. The cells were treated for 72 h with various elicitors, such as MJ, MV, SA, Chi, and ET. MJ and SA were dissolved in ethanol, and MV, Chi, and ET were dissolved in water. The elicitors of MJ, MV, SA, ET, and Chi were used at a final concentration of 100, 10, 100, 300 μM, and 200 μg/mL, respectively. For UV treatment, cell cultures in 250 mL flask were opened in a laminar flow hood and exposed to UV-C light (254 nm, 30 W, 7.3 uW/cm^2^) for 30 min at an irradiation distance of 50 cm, and then the cells were kept for 72 h. The cells were harvested by filtration through a nylon mesh filter for RNA isolation and extraction.

### 3.3. RNA Extraction and Quantitative RT-PCR (qRT-PCR) Analysis

Total RNA isolation, cDNA synthesis, and RT-PCR were performed as previously described [23]. The cDNA was diluted 50-fold, and 1 μL of the diluted cDNA was used in a 20 μL PCR reaction. qRT-PCR was performed using the TransStart Tip Green qPCR Supermix. Assays were performed using a CFX96 real-time PCR detection system (Bio-Rad, CFX96) with thermal parameters of 95 °C for 15 min, followed by 45 cycles at 95 °C for 20 s and 60 °C for 40 s. Transcript levels were calculated after normalization against the actin gene (*SfACT11*) as an internal control. Relative changes in gene expression levels were determined by the 2^−ΔΔCt^ method [47]. Data were analyzed and presented as average values ± standard deviation (*n* = 3). Primers were designed using the Primer3 software [48]. The primers used in this study have been shown in Appendix A.

### 3.4. Statistical Analysis

All the experiments were performed in triplicate, and the data were expressed as the mean ± standard deviation (SD). Subsequent multiple comparisons were analyzed by a Kruskal–Wallis test with Dunn’s multiple comparison post-hoc test. All analyses were performed using Statistical Package for the Social Sciences software version 22 (SPSS Inc., Chicago, IL, USA), and statistical significance was set at *P* < 0.05.

### 3.5. Quantification of Pterocarpans by High-performance Liquid Chromatography (HPLC)

Cell suspension cultures were subcultured in 250 mL flasks containing 50 mL MS1B2P liquid medium and incubated on a rotary shaker (90 rpm) at 24 °C in dark conditions for 7 days. The cells were treated with elicitors for the indicated periods. Pterocarpans were extracted from 0.1 g of the lyophilized powder samples by sonication for 30 min at 45 °C in 8 mL of 100% methanol. After extraction, samples were concentrated using a speed-vac and dissolved in 80% methanol. For analysis of pterocarpans in the cell culture medium, 10 mL of the culture medium was extracted with an equal volume of ethyl acetate, concentrated by evaporation, and dissolved in 1 mL of 80% methanol. HPLC analysis was performed with an Agilent series 1200 quaternary solvent delivery system, cooled auto-sampler (4 °C), and a photodiode array detector (Agilent Technologies). The samples were separated on the Phenomenex synergy polar column (250 mm × 4.6 mm, 4 μM). The column was maintained at 30 °C. The volume of injection was 5 μL. The mobile phase consisted of (A) 0.1% formic acid in water, and (B) 0.1% formic acid in acetonitrile with the following gradient: 0 min, 30% B; 17 min, 70% B; 24 min, 98% B; 29 min, 30% B. The flow rate was 1 mL/min, with detector wavelengths at 310 nm.

### 3.6. Recycling Preparative HPLC and NMR 

The trifolirhizin malonate was separated by the recycling preparative HPLC (LC-9110II NEXT; Japan Analytical Industry Co., Tokyo, Japan). The chromatographic separation was performed on a prepacked column (JAIGEL-ODS-AP-L, 500 mm × 20 mm, 10 μm). Acetonitrile (20%) was used as the mobile phase, and the flow rate was set at 6.0 mL/min. In the preparative HPLC, trifolirhizin malonate was collected between 36 min and 48 min. The solvent was dried by an evaporator and the purified compound was characterized by an NMR. NMR spectra were recorded with a JEOL ECX-500 instrument (^1^H NMR at 500 MHz, ^13^C NMR at 125 MHz; JEOL, Tokyo, Japan) in dimethyl sulfoxide-*d_6_* (*d_6_*-DMSO). The spectra for trifolirhizin malonate is as follows: ^1^H-NMR (500 MHz, DMSO-*d*_6_) δ 7.34 (1H, *d, J* = 9.0 Hz), 6.94 (1H, s), 6.69 (1H, *dd, J* = 2.5, 9.0 Hz), 6.50 (2H, *d, J* = 2.5, 5.0 Hz), 5.91 (2H, *d, J* = 18.5 Hz), 5.51 (1H, *d, J* = 4.0 Hz), 4.83 (1H, *d, J* = 7.5 Hz), 4.24 (1H, *d, J* = 6.0 Hz), 4.09 (2H, m), 3.57~ 3.06 (6H, m), 2.88 (2H, s); ^13^C-NMR (125 MHz, DMSO-*d*_6_) δ 170.135 (C-a), 168.3 (C-b), 147.9 (C-5), 141.6 (C-6), 132.4 (C-7), 118.7 (C-8), 110.7 (C-c), 101.5 (C-d), 93.7 (C-9), 66.4 (C-4), 63.5 (C-3), 46.2 (C-1), 40.1 (C-2).

## Figures and Tables

**Figure 1 ijms-21-03944-f001:**
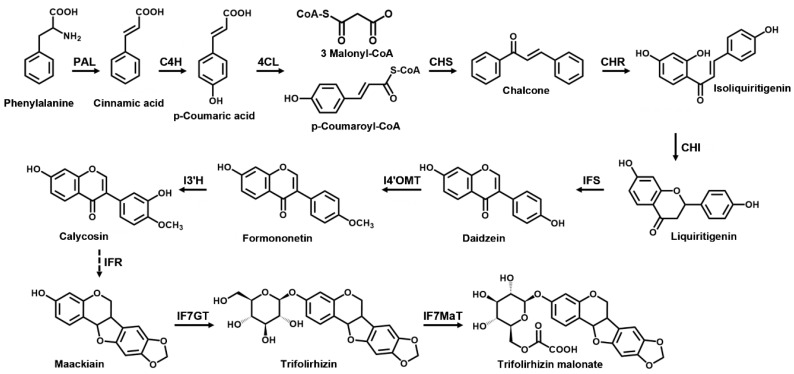
Proposed pterocarpan biosynthetic pathway in *S. flavescens*. The step-wise activity of PAL, C4H, 4CL, CHS, CHR, CHI, IFS, I4’OMT, I3’H, IFR, IF7GT, and IF7MaT result in the conversion of phenylalanine to pterocarpans (maackiain, trifolirhizin, and trifolirhizin malonate). PAL: phenylalanine ammonia-lyase; C4H: cinnamate-4-hydroxylase; 4CL: 4-coumarate-CoA-ligase; CHS: chalcone synthase; CHR: chalcone reductase; CHI: chalcone isomerase; IFS: isoflavone synthase; I4′OMT: isoflavone 4′-*O*-methyltransferase; I3′H: isoflavone 3′-hydroxylase; IFR: isoflavone reductase; IF7GT: UDP-glucose:isoflavone 7-*O*-glucosyltransferase; IF7MaT: malonyl-CoA:isoflavone 7-*O*-glucoside-6-*O*-malonyltransferase. The pterocarpan biosynthetic pathway was proposed based on the KEGG database (http://www.genome.jp/kegg/kegg2.html).

**Figure 2 ijms-21-03944-f002:**
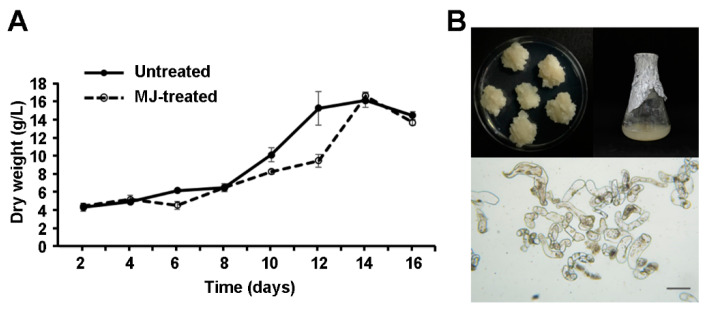
Growth curve analyses of the *S. flavescens* cell cultures. (**A**) The growth of cells treated with or without methyl jasmonate (MJ) in 125 mL flask cultures containing 25 mL of MS1B2P liquid medium was measured at different time points. Data are the mean of three independent replicates ± SD. (**B**) The morphology of *S. flavescens* callus and cell suspension cultures. The cells were observed under an inverted light microscope (Carl Zeiss, Germany) equipped with a camera. Scale bar indicates 100 μm.

**Figure 3 ijms-21-03944-f003:**
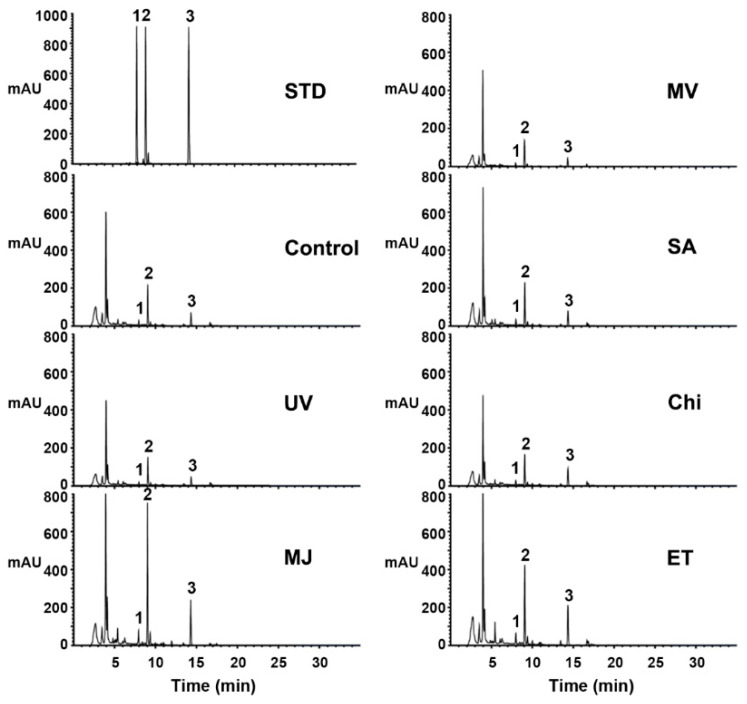
Effects of various elicitors on the production of pterocarpans (maackiain, trifolirhizin, and trifolirhizin malonate) in the *S. flavescens* cell cultures. Cells were pre-cultured for seven days and harvested three days after elicitation with the indicated elicitors. Samples were extracted with methanol and analyzed by HPLC. The elicitors used are as follows: UV, ultraviolet ray; MJ, methyl jasmonate; MV, methyl viologen; SA, salicylic acid; Chi, chitosan; ET, ethephon. Cells treated with 0.1% ethanol was used as a control. Chromatogram STD represents the authentic standards 1 (trifolirhizin), 2 (trifolirhizin malonate), and 3 (maackiain), with retention times of 7.88, 8.93, and 14.28 min, respectively.

**Figure 4 ijms-21-03944-f004:**
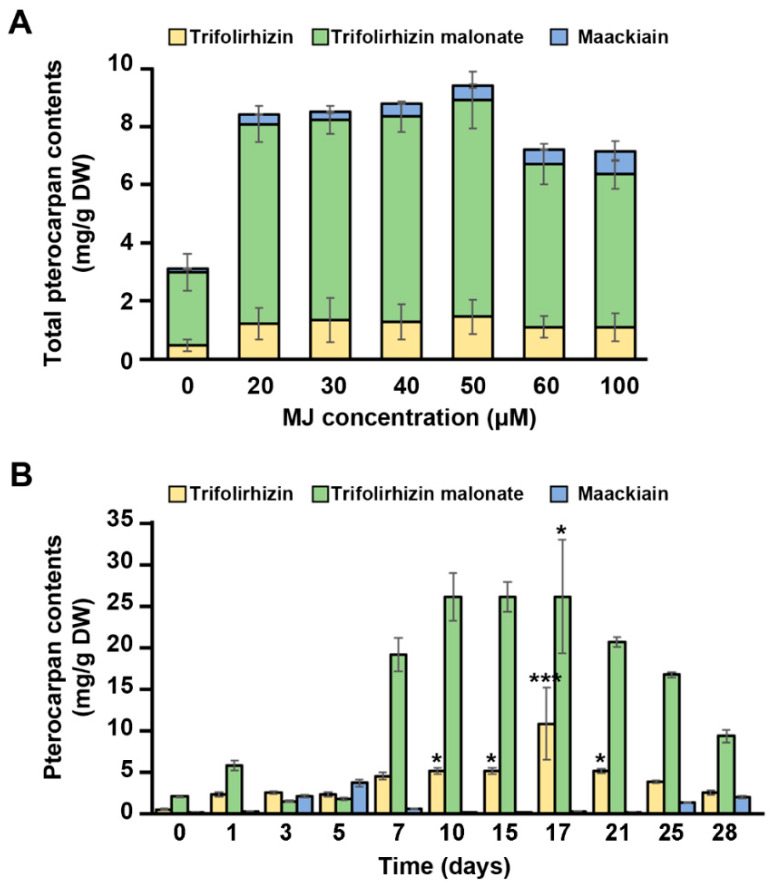
Effects of MJ on the production of pterocarpans (maackiain, trifolirhizin, and trifolirhizin malonate) in *S. flavescens* cell cultures. (**A**) Effects of different concentrations of MJ on pterocarpan production in the *S. flavescens* cell cultures. Cells were pre-cultured for 7 days and harvested 3 days after elicitation with the indicated concentration of MJ. (**B**) Time-course of pterocarpan production in the *S. flavescens* cells elicited with MJ. The cells were pre-cultured for seven days and harvested at the indicated time points after elicitation with 50 μM MJ. Cells treated with 0.05% ethanol were used as a control. Samples were extracted with methanol and analyzed by HPLC. Data are the mean of three independent replicates ± SD. Statistical significance used Kruskal–Wallis test, * *p* < 0.05, *** *p* < 0.001.

**Figure 5 ijms-21-03944-f005:**
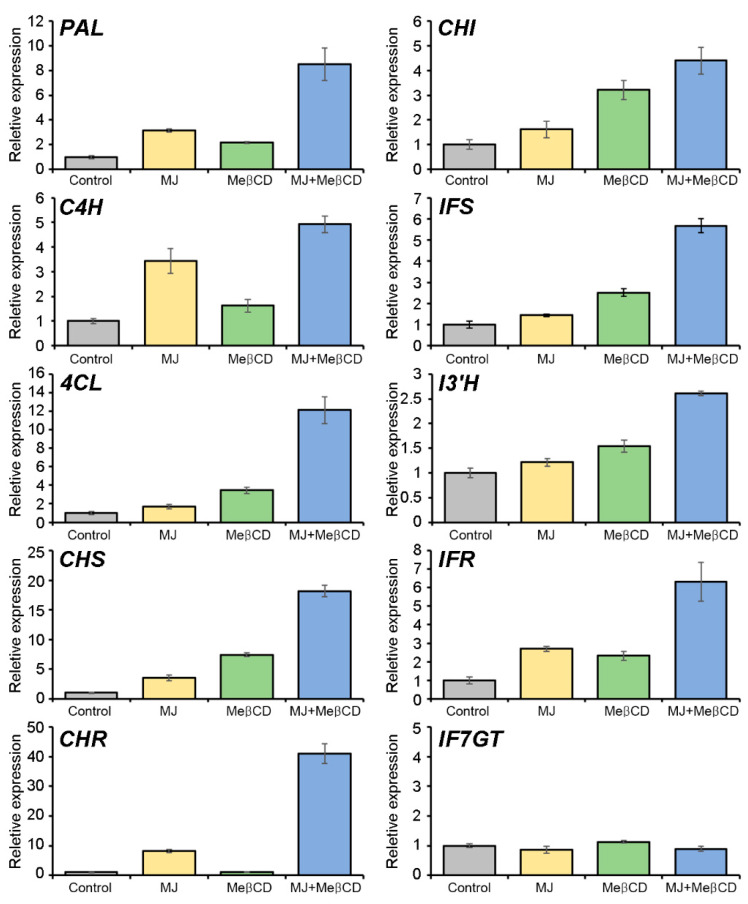
Relative expression of the pterocarpan biosynthetic genes in the *S. flavescens* cells elicited by MJ, methyl-β-cyclodextrin (MeβCD), and MJ + MeβCD. Transcript levels of *PAL, C4H, 4CL, CHS, CHR, CHI, IFS, I3’H, IFR*, and *IF7GT* were analyzed by qRT-PCR, using *S. flavescens* cells elicited with 50 μM MJ, 50 mM MeβCD, or 50 μM MJ + 50 mM MeβCD for 24 h. The relative expression levels were normalized to that of actin (*SfACT11*; transcript ID: *Sf128341_c1_g1_i1*) as a quantitative control, and have been presented as the fold induction relative to the control. Data are the mean of three independent replicates ± SD.

**Figure 6 ijms-21-03944-f006:**
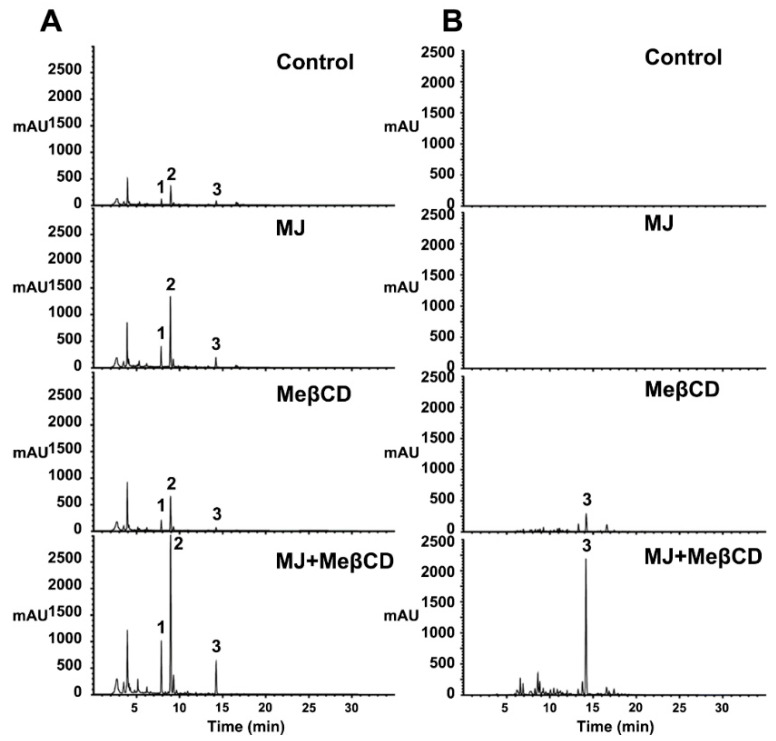
Enhanced production of pterocarpans by MJ and MeβCD co-treatment in the *S. flavescens* cell cultures. (**A**) Synergistic effect of MJ and MeβCD on pterocarpan production in the cells. (**B**) Extracellular production of maackiain in the culture medium. Cells were pre-cultured for 7 days and harvested 7 days after co-treatment with 50 µM MJ and 50 mM MeβCD. Peaks 1 (7.86 min), 2 (8.94 min), and 3 (14.21 min) on the chromatograms represent the peaks identical to the retention times of the authentic standards, trifolirhizin, trifolirhizin malonate, and maackiain, respectively. The culture medium was extracted with ethyl acetate and analyzed by HPLC.

**Table 1 ijms-21-03944-t001:** Time course of pterocarpan production in the cells of MJ-treated *S. flavescens* cell cultures.

		Pterocarpan Contents (mg/g Dry Weight (DW))			
Treatment	Days	Trifolirhizin	Trifolirhizin Malonate	Maackiain	Total
**Control**	0	0.34 ± 0.02	1.65 ± 0.05	0.02 ± 0.00	2.02
	5	0.79 ± 0.04	1.50 ± 0.04	0.06 ± 0.00	2.35
	14	0.71 ± 0.16	1.56 ± 0.43	0.09 ± 0.03	2.36
	25	0.68 ± 0.01	1.65 ± 0.11	0.13 ± 0.01	2.46
	28	0.81 ± 0.02	1.91 ± 0.08	0.08 ± 0.00	2.80
**MJ**	0	0.51 ± 0.12	2.06 ± 0.01	0.11 ± 0.04	2.69
	1	2.35 ± 0.25	5.80 ± 0.61	0.26 ± 0.03	8.41
	3	2.55 ± 0.17	1.46 ± 0.09	2.14 ± 0.15	6.15
	5	2.30 ± 0.25	1.76 ± 0.17	3.69 ± 0.42	7.75
	7	4.54 ± 0.46	19.17 ± 2.05	0.57 ± 0.06	24.28
	10	5.58 ± 0.38	26.66 ± 2.89	0.14 ± 0.01	32.37
	15	5.18 ± 0.39	26.13 ± 1.77	0.11 ± 0.01	31.42
	17	10.81 ± 4.35	26.16 ± 6.86	0.27 ± 0.01	37.24
	21	5.12 ± 0.27	20.68 ± 0.62	0.15 ± 0.62	25.95
	25	3.83 ± 0.15	16.74 ± 0.29	1.36 ± 0.00	21.94
	28	2.49 ± 0.28	9.35 ± 0.75	2.02 ± 0.15	13.86

Cells were pre-cultured for 7 days and harvested at the indicated time points after elicitation with 50 μM MJ. Cells treated with 0.05% ethanol were used as a control. Data are the mean of three independent replicates ± SD.

**Table 2 ijms-21-03944-t002:** Enhanced production of pterocarpans in the cells and medium of *S. flavescens* cell cultures, elicited by MJ, MeβCD, and MJ + MeβCD.

		Pterocarpan Contents			
	Treatment	Trifolirhizin	Trifolirhizin Malonate	Maackiain	Total
**Cells** (mg/g DW)	Control	1.25	4.56	0.31	6.13
	MJ	4.30	17.37	0.75	22.42
	MeβCD	2.21	8.42	0.24	10.86
	MJ + MeβCD	11.64	43.90	2.49	58.04
**Medium** (mg/L)	Control	N.D	N.D	N.D	N.D
	MJ	N.D	N.D	N.D	N.D
	MeβCD	8.55	13.24	30.36	52.16
	MJ + MeβCD	16.73	71.75	222.67	311.16

Cells were pre-cultured for 7 days and harvested 7 days after co-treatment with 50 µM MJ and 50 mM MeβCD.

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
