# Peer review of "The Synergistic Effect of Co-Treatment of Methyl Jasmonate and Cyclodextrins on Pterocarpan Production in Sophora flavescens Cell Cultures"

_ijms, 2020, doi:10.3390/ijms21113944_

Round 1

Reviewer 1 Report

The article by Kim at al. concerns with the effect of co-treatment of MJ and MDC on sophora flavescens cell cultures. The true novelty of the work lies more on the choice of the plant species than the approach itself, but the scientific soundness is good nonetheless and the paper it well written.

Nevertheless there are some minor points to be addressed before publishing.

First of all the authors should insert the full name of the elicitators the first time they are mentioned in the text (line 105).

Line 251: the Authors should add details on NMR spectrometer and experimental parameters

Figure 4: please check the ANOVA significancy. For example in fig 4B, day 3: how can the Maackiain content be statistically different from Trifolirhizin one?

Reviewer 2 Report

Review

The paper entitled “Synergistic effect of co-treatment of methyl jasmonate and cyclodextrins on pterocarpan production in Sophora flavescens cell cultures” is interesting and current. However, there are some points, which should be improved or clarified before publication in Int. J. Mol. Sci. Below there is a list of notes, questions and comments.

  1. Line 19 family Leguminosae is the older name of the family. More appropriate name of the family, which the Authors used in Introduction, is Fabaceae. I suggest using the same name in the whole paper.
  2. Line 19 In my opinion, it will be better to change the sentence into …. Asian medicinal plant. As medicinal plant, is the species (or genus) listed in any Pharmacopoeia?
  3. Line 26 the concentration of MJ is not visible correctly in my proof, the same is with the whole name of MCD in line 27. Is it written correctly?
  4. Lines 50-51 and 61 – different font size
  5. Introduction section: What was the novelty of the paper? What has been reported for the first time in the paper? Were cell cultures of the species reported earlier? Which cultures produced the same metabolites and what was the level of the production? Why pterocarpans are so important? Do they act only as phytoalexins in plants?
  6. Line 91. Is the sentence written correctly? “…during the first 1 day-8 days”?
  7. Result section: In my opinion, some number values should be given in the result description, for example DW in 14 day of cell culture. What was the morphology of the cell suspension? The Authors have written that MJ with MCD used at the same time gave the best results, but I can`t see any figure nor table included to support the results. In my opinion, only chromatograms (Figure 6) are not sufficient in the case, because the readers have not known the values of pterocarpans production after treatment.
  8. Line 95. Have the Authors weighed calli or cell suspension? Which medium?
  9. Discussion is poorly written in some cases and it should be supplemented.
  10. Line 105. There are abbreviations of elicitors used in the experiments, but they are listed for the first time in the line, and readers do not know what the abbreviations mean, because they are clarified in the next part of manuscript. I suggest that for the first time, the whole names of elicitors should be listed in Results section.
  11. Which control conditions were used in the experiments with elicitors? The control conditions should be better indicated in the text and Figure legends. I think that the production of the main three tested metabolites during growth cycle should be also presented on a separate figure, similarly to DW on Figure 2. In my opinion, Fig. 4 is not sufficient, because there is no information what production level is in cell suspension without elicitation. What about DW after elicitation? Was it investigated?
  12. Line 204 In my opinion, the whole name of BAP should be written.
  13. How were the elicitor solutions prepared? Each elicitor in the same way? Which concentrations? What about UV? Was the effect of solvent on the cell culture growth and metabolite production investigated?
  14. Line 227 Statistical analysis section: The given results are means ± SD or SE? There is no information. It should be inserted in the text and figures. The statistical analysis was performed for the each metabolite separately? It should be also listed in the Figure legend. Why was the Tukey test applied? Were the assumptions for the test fulfilled? I suggest to reconsider using of a non-parametric test. I also suggest checking very carefully the letters indicating significance in Figure 4 B), because the lack of differences in some cases looks less probable, for example the lack of differences (the same letter b) in trifolirhizin malonate content between appr. 2 and 16 mg/g DW (????). What about statistical analysis of results presented in Figure 4A? I think that the statistical analysis in the case should be also performed.
  15. In my opinion, a photograph of cell suspension culture used in the experiments should be included.
  16. 2 In my opinion the name of medium should be listed in the Figure legend as well as information whether SD or SE was calculated, as well as how many flasks were used for each repetition (n=?). In my opinion, the sentences: The growth of cells in flask cultures was measured at different time points. “ and “The maximum cell growth was obtained on day 14 of the S. flavescens cell cultures.” are unnecessary, because the information is visible in the Figure.
  17. Figure 5 presents relative expression of the pterocarpan biosynthetic genes in the MJ-treated flavescens cells. Why the Authors have chosen the treatment only with MJ, although they have written that better results gave the co-treatment of cells with MJ and MCD?

To conclude, I think that the results are interesting, but the paper needs Major Revision before publication in Int. J. Mol. Sci.

Round 2

Reviewer 2 Report

The Authors have corrected and improved the manuscript. All my remarks have been considered. However, there are in the text several minor errors which should be corrected before the publication. Below is the list of my remarks:

1.  there is different font size in several lines (lines 1, 59, 70, 137-138, 296-297, 318)

2. line 44 in my opinion, the word "and" should be deleted from the sentence

3. line 97 it should be "flavescens" not "favescens"

4. Fig. 2 A scale bar in the photo should be added

5. Table 1 Control conditions should be indicated under the table. Which concentration of ethanol was used as control conditions (0.1 or 0.05%) 

6. Fig. 4B There are not total pterocarpan contents (the description of axis should be changed)

7. The assumptions for Tukey and Duncan tests are the same. Were they fullfilled? If not, you should used a non-parametric test, which of course can give the other significance, because it is the non-parametric test. 

Author Response

We are happy to hear that you decided to accept our manuscript for publication after major revisions in International Journal of Molecular Sciences.

Our manuscript ID. ijms-784448 entitled "Synergistic effect of co-treatment of methyl jasmonate and cyclodextrins on pterocarpan production in Sophora flavescens cell cultures" has been revised based on your and the reviewers’ comments.

We carefully changed our manuscript in response to each comment and highlighted the changes in the manuscripts by using red text. We are sure that your and the reviewers’ comments should improve quality of our manuscript for publication in International Journal of Molecular Sciences.

Please let us know if we need additional changes in our new version of the manuscript.

Sincerely,

Cha Young Kim, Ph.D.

Tel: +82-63-570-5600

Fax: +82-63-570-5609
